# Bile Acid Sequestration via Colesevelam Reduces Bile Acid Hydrophobicity and Improves Liver Pathology in *Cyp2c70*^−/−^ Mice with a Human-like Bile Acid Composition

**DOI:** 10.3390/biomedicines11092495

**Published:** 2023-09-08

**Authors:** Anna Palmiotti, Hilde D. de Vries, Milaine V. Hovingh, Martijn Koehorst, Niels L. Mulder, Esther Verkade, Melany K. Veentjer, Theo H. van Dijk, Vincent W. Bloks, Rick Havinga, Henkjan J. Verkade, Jan Freark de Boer, Folkert Kuipers

**Affiliations:** 1Department of Pediatrics, University of Groningen, University Medical Center Groningen, 9713 GZ Groningen, The Netherlands; a.palmiotti@umcg.nl (A.P.); m.v.hovingh@umcg.nl (M.V.H.); n.l.mulder@umcg.nl (N.L.M.); e.verkade@umcg.nl (E.V.); m.k.veentjer@umcg.nl (M.K.V.); v.w.bloks@umcg.nl (V.W.B.); h.havinga@umcg.nl (R.H.); h.j.verkade@umcg.nl (H.J.V.); 2Department of Laboratory Medicine, University of Groningen, University Medical Center Groningen, 9713 GZ Groningen, The Netherlands; h.d.de.vries@umcg.nl (H.D.d.V.); t.van.dijk@umcg.nl (T.H.v.D.); 3European Research Institute for the Biology of Ageing (ERIBA), University of Groningen, University Medical Center Groningen, 9713 GZ Groningen, The Netherlands

**Keywords:** bile acids, colesevelam, enterohepatic circulation, liver, humanized mouse model

## Abstract

Bile acids (BAs) and their signaling pathways have been identified as therapeutic targets for liver and metabolic diseases. We generated *Cyp2c70*^−/−^ (KO) mice that were not able to convert chenodeoxycholic acid into rodent-specific muricholic acids (MCAs) and, hence, possessed a more hydrophobic, human-like BA pool. Recently, we have shown that KO mice display cholangiopathic features with the development of liver fibrosis. The aim of this study was to determine whether BA sequestration modulates liver pathology in Western type-diet (WTD)-fed KO mice. The BA sequestrant colesevelam was mixed into the WTD (2% *w*/*w*) of male *Cyp2c70*^+/+^ (WT) and KO mice and the effects were evaluated after 3 weeks of treatment. Colesevelam increased fecal BA excretion in WT and KO mice and reduced the hydrophobicity of biliary BAs in KO mice. Colesevelam ameliorated diet-induced hepatic steatosis in WT mice, whereas KO mice were resistant to diet-induced steatosis and BA sequestration had no additional effects on liver fat content. Total cholesterol concentrations in livers of colesevelam-treated WT and KO mice were significantly lower than those of untreated controls. Of particular note, colesevelam treatment normalized plasma levels of liver damage markers in KO mice and markedly decreased hepatic mRNA levels of fibrogenesis-related genes in KO mice. Lastly, colesevelam did not affect glucose excursions and insulin sensitivity in WT or KO mice. Our data show that BA sequestration ameliorates liver pathology in *Cyp2c70*^−/−^ mice with a human-like bile acid composition without affecting insulin sensitivity.

## 1. Introduction

In addition to their classical function in the absorption of dietary fat-soluble nutrients and vitamins, bile acids (BAs) have been identified as signaling molecules involved in the regulation of lipid, glucose and energy metabolism and the modulation of immune responses [1]. Their action on metabolism is manifested through the activation of several intracellular nuclear receptors, such as the farnesoid X receptor (FXR/NR1H4), pregnane X receptor (PXR/NR1I2), vitamin D receptor (VDR) and the constitutive androstane receptor (CAR/NR1I3), as well as cell surface G protein-coupled receptors (GPCRs), such as the Takeda G protein-coupled receptor 5 (TGR5/GPBAR1). Due to their involvement in multiple metabolic pathways, BAs and their signaling pathways represent targets for therapeutic intervention in (metabolic) diseases. In fact, BAs have been used for many years to treat gallstones and cholestatic liver diseases [2], and currently ursodeoxycholic acid (UDCA) is part of the standard treatment for primary biliary cholangitis (PBC) [3]. Additionally, synthetic and natural modulators of FXR have been tested in animal models displaying liver inflammation, fibrosis and liver steatosis [4,5]. In addition, beneficial effects of pharmacological FXR stimulation have also been demonstrated in clinical trials: Obeticholic acid (OCA) treatment has been shown to exert positive effects in patients with primary biliary cholangitis (PBC) [6,7,8] and is now approved by the FDA and EMA for the treatment of PBC in patients that have an incomplete response or are intolerant to UDCA [9]. Furthermore, it is well-established that BA sequestrants, compounds that bind BAs in the intestine and thereby prevent their re-absorption, reduce plasma LDL cholesterol levels and cardiovascular diseases in patients with hyperlipidemia through stimulating BA production and, thus, cholesterol catabolism [10]. Additionally, inhibiting intestinal BA re-absorption (either with BA sequestrants or through pharmacological ASBT inhibition) ameliorates BA-mediated cholestatic liver and bile duct injury in mouse models of cholestatic liver injury and sclerosing cholangitis [11,12]. However, studies have also indicated that the BA sequestrant colesevelam may lower plasma glucose levels as wells as HbA1c in patients with type 2 diabetes mellitus (T2DM) [13,14,15]. BA sequestration reduces plasma glucose levels in diabetic *db*/*db* mice by increasing the metabolic clearance rate in peripheral tissues [16]. However, the exact mechanism driving the improved glucose homeostasis upon BA sequestration remains to be elucidated. On the other hand, it has been reported that BA sequestration may increase VLDL-triglyceride (TG) production and results in moderately elevated plasma TG levels in humans [17,18]. The majority of preclinical studies that have provided the basis for the current understanding of the impact of BAs and their signaling pathways on metabolism and liver function have been conducted in murine models. However, several differences in BA metabolism between mice and humans complicate the translation of murine experimental data to the human situation. In contrast to humans, mice metabolize chenodeoxycholic acid (CDCA) to muricholic acids (MCAs) in the liver, which is mediated by the mouse-/rat-specific enzyme CYP2C70. Therefore, hydrophilic and cytoprotective MCA species, which are poor agonists for FXR and TGR5, are highly abundant in the murine BA pool, while the most potent physiological agonist of the FXR, hydrophobic and cytotoxic CDCA, is abundant in the human bile acid pool but only present in very low amounts in mice. To better translate murine data to the human situation, a mouse model with a human-like BA metabolism has been developed [19,20,21]. In this model, the gene encoding CYP2C70 has been inactivated [19]. It has been reported that *Cyp2c70*^−/−^ mice show elevated plasma bile acid and transaminase levels and develop hepatic fibrosis [19,22]. In this study, we aimed to investigate if BA sequestration impacts hepatic lipid accumulation and other liver-related pathological features in WTD-fed KO mice.

## 2. Materials and Methods

### 2.1. Animals

The 12- to 14-week-old male *Cyp2c70*^−/−^ (KO) mice (n = 17) on a C57BL/6J background (C57BL/6J-Cyp2c70^em3Umcg^) and wild-type (WT) littermates (n = 18), bred at the local animal facility (CDP) of the University Medical Center Groningen, were housed individually in a temperature-controlled room (21 °C) with a light/dark-cycle of 12 h each. During the study, mice had ad libitum access to food and water. Mice were exposed to a Western-type diet (WTD), containing 60% energy from fat and 0.25% added cholesterol (D14010701Bi, Research Diets Inc., New Brunswick, NJ, USA), for a period of 8 weeks prior to colesevelam intervention. Some of the mice (KO mice n = 9, WT mice n = 9) were subsequently treated with 2% (*w*/*w*) colesevelam hydrochloride (HCl) (182815-44-7 Daiichi Sankyo Pharma Development, Edison, NJ, USA) mixed into the WTD, while the other animals (KO mice n = 8, WT mice n = 9) remained on WTD without colesevelam as the control. Body weight (BW) and food intake (FI) were monitored weekly. Body composition was analyzed after 2 weeks of colesevelam treatment. Oral glucose tolerance tests and insulin analysis were performed at 2 weeks after start of the colesevelam treatment. After 3 weeks of colesevelam treatment, mice were euthanized under isoflurane anesthesia through cardiac puncture followed by cervical dislocation. Plasma and organs were collected and stored at −80 °C until analysis. Fecal samples were collected at the end of colesevelam treatment for 3 days prior to termination. All animal experiments were approved by the Dutch Central Committee for Animal Experiments and the Animal Welfare Body of the University of Groningen (Dutch National Central Commission for Animal Experiments, CCD, license number AVD10500202115290).

### 2.2. Body Weight, Food Intake and Body Composition Analysis

To determine body weights and food intake, weekly measurements were made on individually housed animals. Using a Minispec Body Composition Analyzer (LF90-II, Bruker BioSpin GmbH, Rheinstetten, Germany), after 2 weeks of treatment, non-invasive measurements of body composition (fat mass, lean mass and fluid mass) were performed on conscious mice after 4 h of fasting.

### 2.3. Assessment of Glucose Tolerance and Insulin Resistance

To examine glucose excursions, an oral glucose tolerance test (OGTT) was performed after 2 weeks of colesevelam treatment. After 4 h of fasting (08:00–12:00 h), mice received an oral glucose bolus (2 g/kg BW) via gavage. Using a portable glucose meter (Accu-Chek Performa, Roche Diabetes Care, Almere, The Netherlands), blood glucose levels were measured before and 5, 15, 30, 45, 60, 90 and 120 min after gavage. To explore the effect of administered glucose on insulin levels, blood spots were also taken before and at 5, 15, 60 and 120 min after gavage for insulin measurement, as previously described [23]. Blood spots were briefly air-dried and stored at −20 °C for insulin measurements (Crystal Chem rat insulin ELISA (#90010) with mouse insulin standard (#90020), Zaandam, The Netherlands). Plasma HOMA-IR index was calculated, multiplying the fasting blood glucose levels by the fasting plasma insulin levels and dividing the product by 14.1.

### 2.4. Plasma Biochemistry

The determination of plasma triglycerides and total cholesterol was performed using commercially available kit reagents: Roche Diagnostics, Rotkreuz, Switzerland and DiaSysDiagnostic Systems, Holzheim, Germany, respectively. The concentrations of the enzymes aspartate aminotransferase (AST), alanine aminotransferase (ALT) and albumin in plasma were determined with a clinical chemistry analyzer (Cobas 6000, Roche Diagnostics) and standard reagents (Roche Diagnostics).

### 2.5. Bile Acid Measurements

Plasma and gallbladder bile BA species were quantified with a liquid chromatography–tandem mass spectrometry technique (LCMS) using a Nexera X2 Ultra High-Performance Liquid Chromatography system (SHIMADZU, Kyoto, Japan) linked to a SCIEX QTRAP 4500 MD triple quadrupole mass spectrometer (SCIEX, Framingham, MA, USA) (UHPLC-MS/MS), as previously described [24]. Total BA concentrations in gallbladder bile were not quantitated since the gallbladder volumes were too small for a precise and quantitative analysis. However, the BA percentages were calculated accurately and used to calculate the hydrophobicity index. To quantify fecal BAs, individual mouse feces were collected for 72 h, then desiccated and finely ground. About 30 mg of dried feces were incubated in 0.5 M sodium hydroxide (NaOH) and methanol (MeOH) at 80 °C for 3 h. The boiled feces were ultra-sonicated and centrifuged. Part of the supernatant was mixed with the internal standard, MeOH and 0.375 M hydrochloric acid (HCL). BAs were purified using Oasis HLB columns (Waters, Milford, MA, USA) rinsed with 0.25 M HCL solution and hexane. Samples were then dried at 50 °C under a stream of nitrogen and dissolved in 50% MeOH for quantification via LCMS using 5β-cholanic acid 7α,12α diol as the internal standard.

### 2.6. Hepatic Lipid Analyses

After lipid extraction from homogenates as described by Bligh and Dyer [25], hepatic triglycerides, total cholesterol, and phospholipids were measured with commercially available reagents (systems from Roche Diagnostics and Diasys Diagnostic, respectively).

### 2.7. Fecal Neutral Sterol Analyses

Individual mouse feces were collected at the end of the treatment, prior the termination, for 72 h, desiccated, weighted and finely ground for the quantification of fecal neutral sterol excretion. For 2 h, 50 mg of dried feces was incubated in alkaline methanol at 80 °C. The neutral sterols were then extracted three times with petroleum ether and derivatized for one hour with pyridine/N,O-Bis (trimethylsilyl) trifluoroacetamide (BSTFA)/trimethylchlorosilane (TMCS) (following the ratio 50:50:1). After drying at room temperature under a nitrogen stream, samples were dissolved in heptane with 1% BSTFA and quantified with GC using 5-cholestane as an internal standard.

### 2.8. Determination of mRNA Levels

TRI reagent (Sigma, St. Louis, MO, USA), NanoDrop (NanoDrop Technologies, Wilmington, DE, USA), and Moloney-Murine Leukemia Virus reverse transcriptase (Life Technologies, Bleiswijk, The Netherlands) were used to extract and reverse transcribe total RNA from the liver and distal small intestine, respectively. The RNeasy Lipid Tissue Mini Kit (QI-AGEN Sciences, Germantown, MD, USA) was used to extract total RNA from brown adipose tissue (BAT), which was subsequently quantified via NanoDrop and reverse transcribed as described previously. On a StepOnePlusTM Real-Time PCR system (Applied Biosystems, Foster City, CA, USA), real-time quantitative PCR analyses were performed. Cyclophilin and 36b4 (Rplp0) were used as housekeeping genes for liver and intestine (Cyclophilin) and BAT (36b4). Gene expression levels were first normalized to the housekeeping genes and then to the mean of the corresponding control group.

### 2.9. Histology and Staining of Liver

After termination, part of the liver tissue was quickly excised and fixed in 4% formalin for 24 h prior to embedding in paraffin. Sections (4 µm) were cut and stained with hematoxylin and eosin to determine liver morphology or Sirius Red/Fast Green to visualize collagen deposition. Immunohistochemical staining for cholangiocytes was performed using an anti-cytokeratin 19 (CK19) antibody (ab52625; Abcam, Cambridge, United Kingdom) in accordance with the manufacturer’s guidelines. Images were obtained using a Hamamatsu NanoZoomer (Hamamatsu Photonics, Almere, The Netherlands).

### 2.10. Determination of Fecal Energy Content and Energy Absorption

Samples of ~300 mg of powdered dry feces were combusted in a Parr 6100 compensated calorimeter (Parr Instrument Company, Moline, IL, USA) with a 1108 Oxygen Bomb placed in 2000 g of demineralized water. The temperature increase of the water determined the caloric content of the feces. The intra-assay variability was ~0.3%. The energy absorption efficiency (%) was calculated as: (1 − (energy loss/energy intake)) × 100 [22].

### 2.11. Statistical Analysis

GraphPad Prism (version 8, GraphPad Software, San Diego, CA, USA) was used to generate Tukey box–whisker plots or line graphs with mean and ±SD to represent all results. Brightstat’s Kruskal–Wallis H test and Conover post hoc comparisons were used to identify significant differences in multiple group comparisons [26]. Differences were regarded as statistically significant when their *p*-values were 0.05 or lower.

## 3. Results

### 3.1. Colesevelam Does Not Affect the Body Weight of WTD-Fed Cyp2c70^−/−^ Mice

Twelve- to fourteen-week-old male Cyp2c70^+/+^ (WT) and Cyp2c70^−/−^ (KO) mice were fed a Western-type diet (WTD) for 8 weeks in order to induce obesity and insulin resistance. Prior to the start of the intervention, the body weights and food intake of WT and KO were comparable during the 8 weeks of WTD feeding (data not shown). Some of the mice were subsequently treated with colesevelam (2% *w*/*w* mixed into the WTD), while the other animals were maintained on the WTD without additions. The average food intake during 3 weeks of treatment, as well as body weights, fat mass and lean mass at the end of treatment, were unaffected by colesevelam treatment (Figure 1A–D).

### 3.2. Colesevelam Reduces the Hydrophobicity of the Bile Acid Pool in WTD-Fed Cyp2c70^−/−^ Mice

Colesevelam, being a BA sequestrant, binds BA in the intestine, leading to the formation of non-absorbable complexes that are excreted into the feces and thereby increases fecal BA loss. In line with this, BA sequestration significantly increased fecal bile acid excretion in both WT and KO mice, indicating that colesevelam indeed interfered with intestinal BA absorption in these mice (Figure 2A). Next, gallbladder bile, plasma BA concentrations and composition were analyzed. As already demonstrated in our previous studies, Cyp2c70 deficiency resulted in the complete absence of hydrophilic murine-specific muricholic acids (MCAs) and high relative abundances of hydrophobic chenodeoxycholic acid (CDCA) in bile and plasma (Figure 2B,D). Consequently, the absence of Cyp2c70 caused a more hydrophobic BA pool, as indicated by a higher hydrophobicity index of biliary BA (Figure 2C). Colesevelam significantly reduced the hydrophobicity index of biliary BAs in KO mice (Figure 2C) due to a relative increase in the proportion of TCA at the expense of TCDCA (Figure 2B). Interestingly, colesevelam had an opposite effect on BA hydrophobicity in WT mice because very hydrophilic TMCAs and TUDCA were relatively lower abundant in bile upon treatment and the proportion of less hydrophilic TCA increased in these mice (Figure 2C,D). Plasma total BA concentrations in KO mice remained unchanged upon colesevelam treatment (Figure 2E). In association with the reduced amounts of non 12α-hydroxylated (T)CDCA and increased proportions of 12α-hydroxylated (T)CA in KO mice treated with colesevelam, their plasma ratio of 12α-/non-12α-hydroxylated BAs was markedly increased compared to untreated KO mice (Figure 2F). Colesevelam had a similar, albeit less pronounced effect on 12α-/non-12α-hydroxylated BA ratios in WT mice. Furthermore, to compensate for the colesevelam-induced fecal BA loss, BA synthesis was stimulated. In fact, hepatic mRNA levels of genes encoding BA synthesis enzymes Cyp7a1, considering the rate-limiting enzyme in bile acid synthesis, and Cyp8b1, controlling for CA synthesis, were significantly increased in colesevelam-treated mice compared to untreated controls (Figure 2G), in line with the shift towards more 12α-hydroxylated (T)CA and less hydrophobic (T) CDCA. The nuclear receptor FXR, once activated, regulates Cyp8b1 and Cyp7a1 by inhibiting their expression. Upon colesevelam treatment, with the sequestration of CDCA, a potent endogenous FXR activator, the activation of FXR may be less effective. However, the hepatic mRNA expression of Fxr (Nr1h4) remained unchanged upon treatment, while the expression of the FXR target Shp (Nr0b2) was significantly reduced in WT mice only upon colesevelam treatment (Figure 2G). Next, we analyzed the mRNA levels of Fxr, Shp and Fgf15 in the terminal ileum. Fxr expression was increased in colesevelam-treated WT mice compared to untreated WT mice, but no significant changes were observed in KO mice upon colesevelam treatment. In line with the effective prevention of intestinal BA uptake, the expression of the FXR downstream target genes Shp and Fgf15 was strongly reduced upon colesevelam treatment in both WT and KO mice (Figure 2H).

### 3.3. Colesevelam Differentially Modulates WTD-Induced Hepatic Steatosis in WT and Cyp2c70^−/−^ Mice

Liver weights were unaffected by colesevelam treatment in WT and KO mice (Figure 3A). Hematoxylin and eosin (H&E) staining of liver sections revealed evidence of hepatic steatosis in WTD-fed WT mice and to a lesser extent in WTD-fed KO mice. Colesevelam treatment ameliorated hepatic steatosis in WT mice, but such an effect was not clearly discernible in KO mice (Figure 3B). Hepatic total cholesterol concentrations were significantly decreased in colesevelam-treated WT and KO mice as compared to untreated controls (Figure 3C). Hepatic phospholipid contents were unaffected by genotype and treatment (Figure 3C). While hepatic tryglyceride (TG) contents were markedly decreased in WT mice upon colesevelam treatment, BA sequestration did not further reduce the already relatively low TG accumulation in livers of KO mice (Figure 3C). Moreover, plasma concentrations of total cholesterol and triglycerides in WT and KO mice were not affected by 3 weeks of BA sequestration (Figure 3D). Surprisingly, colesevelam treatment had no impact on the hepatic mRNA levels of LDL receptor (Ldlr) (Figure 3E). Yet, there was a significant increase in the hepatic mRNA levels of the gene-involved cholesterol biosynthesis (Hmgcr) in colesevelam-treated mice compared to untreated mice of both genotypes (Figure 3E), indicating that treatment with the BA sequestrant indeed induced cholesterol synthesis to provide substrate (cholesterol) for BA production. Taken together, these results suggest that cholesterol synthesis, and not the uptake of cholesterol-carrying lipoprotein particles, is stimulated by colesevelam. The mechanisms underlying the absence of effects on Ldlr expression, however, remain to be elucidated. Hepatic mRNA levels of Nr1h3 (Lxra), an important regulator of cholesterol and lipid metabolism, as well as expression levels of the lipogenic genes, Acc1 (Acaca), Fasn and Scd1, were unchanged in KO mice upon colesevelam treatment, while only Srebp1c transcript levels were significantly reduced in WT mice treated with colesevelam as compared to untreated controls (Figure 3F). Moreover, colesevelam treatment significantly increased hepatic mRNA levels of Mttp, encoding a protein involved in the VLDL assembly in KO mice (Figure 3G), but it did not affect hepatic mRNA levels of Tm6sf2, a gene involved in the VLDL secretion (Figure 3G). BA sequestration did not seem to affect fatty acid oxidation, as the expression of Ppara, Cpt1a and Acox1 remained unaffected by colesevelam treatment in WTD-fed WT as well as in KO mice (Figure 3H).

### 3.4. Colesevelam Ameliorates Liver Damage in WTD-Fed Cyp2c70^−/−^ Mice

Liver damage markers AST and ALT in plasma of WTD-fed KO mice that did not receive colesevelam treatment were significantly higher than in WT littermates. Importantly, 3 weeks of colesevelam treatment fully normalized ALT and AST levels in KO mice (Figure 4A). Next, liver sections were stained with Sirius Red to visualize collagen deposition. In accordance with our previous findings, livers of untreated WTD-fed KO mice showed moderately increased collagen deposition (stage F2) compared to their WT littermates (stage F0/F1), which was mainly localized in the periportal areas. However, considerable variation in the degree of fibrosis was observed within the group of untreated KO mice (Figure 4B). Three weeks of treatment with colesevelam did not translate into a significant reduction in the collagen-stained areas in livers of KO mice (Figure 4B). Interestingly, hepatic mRNA levels of genes involved in fibrogenesis were elevated in untreated KO mice compared to WT and markedly upon colesevelam treatment (Figure 4C), suggesting that changes in fibrogenesis are apparent, but that these have not yet translated into detectable differences in the amounts of deposited collagen in the livers of KO mice within the 3-week timeframe of treatment that was applied. Moreover, the expression of monocyte chemoattractant protein-1 (Mcp1, Ccl2) was significantly decreased in colesevelam-treated KO mice as compared to untreated KO mice (Figure 4D). However, the hepatic expression of the macrophage marker F4/80 (Adgre1) did not differ between any of the groups. Additionally, the expression of the marker of cellular senescence P16 (cdkn2a) was significantly increased in untreated KO mice as compared to WT, but was not decreased upon colesevelam treatment (Figure 4D). CK19 immuno-staining of liver sections revealed a significant increase in the number of cholangiocytes in the livers of untreated WTD-fed KO mice compared to WT controls. However, the reduction in the amount of cholangiocytes in livers that was achieved with 3 weeks of colesevelam treatment in KO mice did not reach statistical significance (Figure 4E).

### 3.5. Colesevelam Modulates the Absorption Efficiency of Dietary Energy in WTD-Fed WT and Cyp2c70^−/−^ Mice

Fecal excretion of neutral sterols (cholesterol and its bacterial derivatives) was increased in untreated WTD-fed KO mice compared to WT controls, which is in line with previous observations [22]. Interestingly, colesevelam treatment induced an increased excretion of fecal neutral sterols in WT mice, whereas such an effect was not discernible upon treatment of KO mice (Figure 5A). In addition, colesevelam-treated WT and KO mice lost more energy in the feces as compared to their untreated controls due to a reduced absorption efficiency of their dietary energy (Figure 5B,C).

### 3.6. Colesevelam Does Not Affect Glucose Excursions and Insulin Sensitivity in WTD-Fed Cyp2c70^−/−^ Mice upon 2 Weeks of Treatment

To evaluate the effects of bile acid sequestration via colesevelam on glucose metabolism and insulin sensitivity, fasting blood glucose and plasma insulin concentrations were determined after two weeks of treatment. Fasting glucose and fasting insulin levels did not, however, differ significantly between any of the groups (Figure 6A). The HOMA-IR was significantly lower in untreated WTD-fed KO as compared to WT controls but was not significantly affected by colesevelam treatment in WT and in KO mice (Figure 6A). Moreover, bile acid sequestration had no significant effects on blood glucose excursions, nor on plasma insulin levels in WT and KO mice during an OGTT (Figure 6B).

## 4. Discussion

In this study, we show that bile acid sequestration via colesevelam increases the proportion of 12α-hydroxylated BAs in the circulating bile acid pool, thereby differentially affecting its hydrophobicity in WT and *Cyp2c70*-deficient mice. Whereas BA sequestration increases the hydrophobicity of biliary BAs in WT mice, it causes a substantial decrease in BA hydrophobicity bile in KO mice. Moreover, our findings provide evidence that colesevelam alleviates hepatic injury in WTD-fed *Cyp2c70*^−/−^ mice without affecting insulin sensitivity in these mice with a human-like bile acid metabolism. As previously reported, by us [19,27] and others [20,21], *Cyp2c70*^−/−^ mice have a human-like bile acid composition, lacking the hydrophilic and hepatoprotective mouse/rat-specific MCA species and showing high abundances of hydrophobic and cytotoxic CDCA in their circulating BA pool. The altered BA composition in *Cyp2c70*^−/−^ mice is associated with the development of cholangiopathy and liver fibrosis [19]. We hypothesize that the beneficial effects of colesevelam on liver pathology in KO mice might be mediated at least in part by a shift in BA production towards more 12α-hydroxylated CA and less CDCA due to increased activity of sterol 12α-hydroxylase in the levers of these mice, resulting in a more hydrophilic, less cytotoxic BA composition in mice lacking *Cyp2c70*. In line with the increased 12α-/non-12α-hydroxylated BA ratio in KO mice following colesevelam treatment, these mice showed a strongly upregulated hepatic mRNA expression of *Cyp8b1*, encoding sterol 12α-hydroxylase. While this manuscript was in preparation, Truong et al. reported that interrupting the enterohepatic circulation of BAs through pharmacological inhibition of the ileal BA transporter (IBAT/ASBT/SLC10A2) improves cholangiopathy in *Cyp2c70*^−/−^ mice [28]. Although the results of that study largely support our conclusions, some differences between the study of Truong et al. and our current study do exist. First of all, Truong and colleagues started their intervention in mice at the age of only 4 weeks and continued for 8 weeks. As we have previously shown [19], *Cyp2c70*^−/−^ do have increased plasma transaminases, indicative of hepatocyte damage, at the age of 3 weeks but do not yet display an increase in hepatic fibrosis. Therefore, with respect to liver fibrosis, the ASBT inhibition as applied by Truong in all likelihood prevented collagen deposition rather than restoring pre-existing fibrosis. In our current study, mice received the colesevelam treatment starting at the age of 20-22 weeks, thus at an age when fibrosis is already evident. Furthermore, in the current study we only applied BA sequestration for 3 weeks, a duration that is considerably shorter compared to the 8 weeks of ASBT inhibition applied by Truong and colleagues. Another difference between our study and the study of Truong was the diet that was given to the mice during the intervention. In the study by Truong and co-workers, the mice were fed a regular chow diet, whereas we fed the mice a WTD in this study in order to explore the metabolic effects of BA sequestration in mice with a human-like BA composition. As male mice are more prone to develop obesity and insulin resistance upon WTD feeding, only male mice were used in our current study. Male *Cyp2c70*^−/−^ mice do, however, display less severe liver pathology compared to females [19,28]. Importantly, a disruption of the enterohepatic circulation of BAs via ASBT inhibition might exert other effects than when using BA sequestrants. BA sequestrants like colesevelam bind BAs in the intestine, preventing their re-absorption and removing them from enterohepatic circulation, whereas pharmacological ASBT inhibition prevents BA re-absorption by blocking the enterocytic uptake transporter, leading to increased amounts of free BAs entering the colon, where they can be deconjugated and converted into secondary species by the bacteria that are populating the colon and subsequently be absorbed by passive diffusion. In line with this, Truong and co-workers reported a substantial increase in the proportion of TDCA in livers of *Cyp2c70*^−/−^ mice upon ASBT inhibition, leading to an increased BA hydrophobicity despite increased 12α-/non-12α-hydroxylated BA ratios. We did not observe such an increase in the proportion of TDCA in bile, but colesevelam treatment did elicit an increase in the 12α-/non-12α-hydroxylated BA ratios, resulting in a considerably reduced hydrophobicity of biliary BAs. Intriguingly, despite the increased BA hydrophobicity in their study, Truong and colleagues did observe robust hepatoprotective effects in *Cyp2c70*^−/−^ upon ASBT inhibition, which were attributed to decreased hepatic total BA concentrations. It is, however, not clear to what extent the reduced total BA concentrations reflect lower BA concentrations within the hepatocytes or whether they reflect an altered contribution of BAs present within the biliary tree in that study. Colesevelam treatment reduced the absorption efficiency of energy from the diet in WT, as well as in KO mice. Surprisingly, this did not translate into lower body weight or decreased adiposity in the mice. We hypothesize that the colesevelam-induced reduction in the hydrophobicity BA pool would lead to the reduced activation of TGR5 in brown adipose tissue (BAT), because more hydrophobic BAs are potent ligands for this receptor. Reduced TGR5 activation would, thus, lead to a reduced amount of energy used for thermogenesis. Energy expenditure and body temperature were not measured in the current study, but we did analyze the mRNA levels of the thermogenic genes *Ucp1* and *Dio2* in BAT. The expression levels of both of these genes were similar in all groups (Appendix A), suggesting that *Cyp2c70* deficiency, as well as the altered BA composition upon colesevelam treatment in these KO mice and their WT littermates, did not impact TGR5 signaling or thermogenesis. The BA receptor FXR is known to regulate glucose metabolism [2] and the FXR agonist OCA, an analogue of CDCA, has been demonstrated to improve insulin sensitivity in patients with type 2 diabetes mellitus [29]. Furthermore, we have previously shown that 2 weeks of colesevelam treatment increases the metabolic clearance rate of glucose in diabetic *db*/*db* mice by improving insulin sensitivity in peripheral tissues [16], and also that CDCA may improve skeletal muscle insulin sensitivity [30]. Therefore, we investigated the effects of 2 weeks of colesevelam treatment on glucose metabolism in mice in the context of a human-like BA composition, i.e., in *Cyp2c70*^−/−^ mice. KO mice had a lower HOMA-IR compared to their WT littermates, but 2 weeks of colesevelam treatment had no effects on HOMA-IR in both KO and WT mice and did not impact glucose excursions and plasma insulin levels during an OGTT. Together, these data indicate that, in contrast to previous observations in *db*/*db* mice [16], colesevelam did not impact insulin sensitivity in WTD-fed *Cyp2c70*^−/−^ mice and in their WT littermates. It can be hypothesized that the unobserved effects (improvement) on glucose homeostasis and insulin sensitivity in our *Cyp2c70*^−/−^ mice could be due to a high sequestration of hydrophobic CDCA via colesevelam. It is known that more hydrophobic BAs are potent agonists for the TGR5 receptor; its activation leads to an improvement of the whole-body glucose homeostasis and insulin sensitivity through the promotion of GLP-1 secretion. Therefore, reduced CDCA levels could not effectively activate this receptor and, hence, may not lead to an improvement in glucose homeostasis. Additionally, decreased CDCA levels may interfere with AKT phosphorylation and lower muscle insulin sensitivity. As insulin resistance in *db*/*db* mice is much more extreme than in WTD-fed C57BL/6J mice, it may be interesting to study the effects of colesevelam in mice with a human-like BA composition in the context of a more severe insulin resistant phenotype at the start of the intervention.

## 5. Conclusions

In conclusion, bile acid sequestration with colesevelam improved liver pathology in *Cyp2c70*^−/−^ mice. The improvement was associated with a reduced hydrophobicity of biliary BAs in *Cyp2c70*^−/−^ mice. However, 2 weeks of BA sequestration had no effect on insulin sensitivity in WTD-fed *Cyp2c70*^−/−^ mice.

## Figures and Tables

**Figure 1 biomedicines-11-02495-f001:**
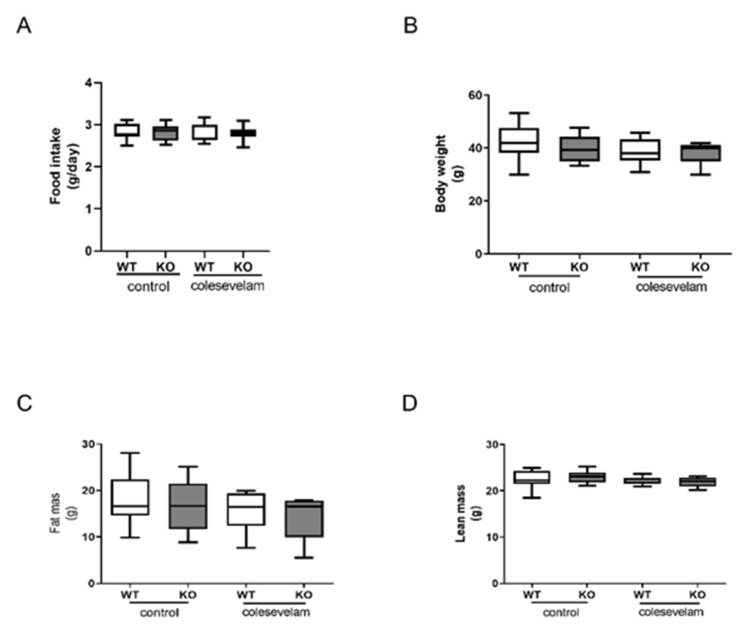
Colesevelam does not affect body weight in WTD-fed male Cyp2c70^−/−^ mice. (**A**) Average food intake and (**B**) body weights of male Cyp2c70^−/−^ mice and WT littermates on control diet and during 3 weeks of colesevelam treatment. (**C**) Fat mass and (**D**) lean mass in Cyp2c70^−/−^ mice and WT littermates on control diet and after 2 weeks of colesevelam treatment, as described in the materials and methods section. N = 8–9 mice/group. Data are presented as Tukey’s box and whisker plots. WT: wild type; KO: Cyp2c70^−/−^, WTD: Western-type diet.

**Figure 2 biomedicines-11-02495-f002:**
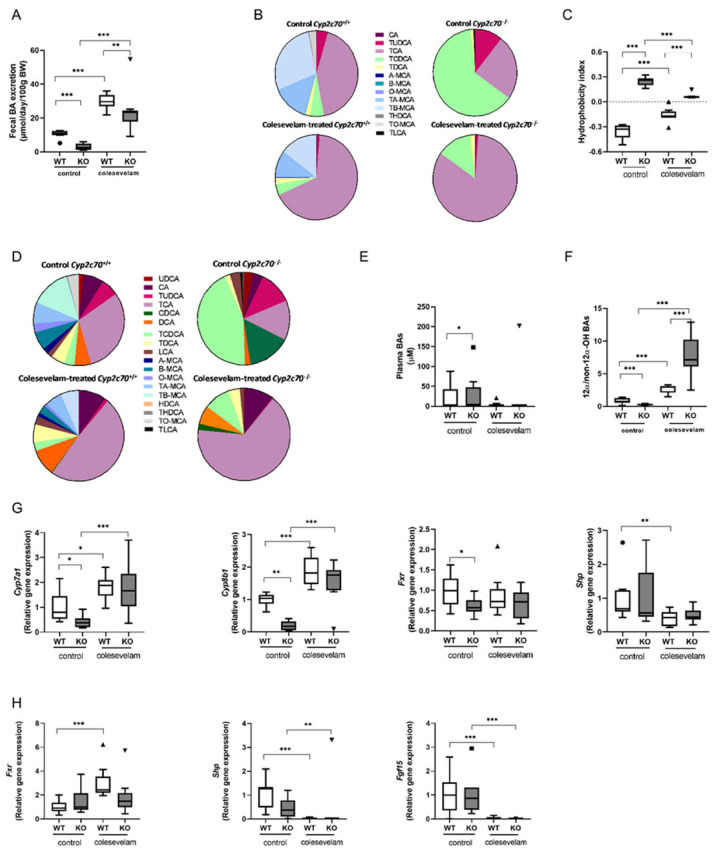
Colesevelam reduces the hydrophobicity of the bile acid pool in Cyp2c70^−/−^ mice. (**A**) Fecal BA excretion in Cyp2c70^−/−^ mice and WT littermates on control diet and after 3 weeks of colesevelam treatment. (**B**) BA composition in gallbladder bile of Cyp2c70^−/−^ mice and WT littermates on control diet and after 3 weeks of colesevelam treatment. (**C**) Hydrophobicity index of biliary bile acids. (**D**) Plasma bile acid composition in Cyp2c70^−/−^ mice and WT littermates on control diet and after 3 weeks of colesevelam treatment. (**E**) Total plasma BA concentrations. (**F**) Plasma ratio 12α-/non-12α-hydroxylated BAs in Cyp2c70^−/−^ mice and WT littermates on control diet and after 3 weeks of colesevelam treatment. (**G**) Hepatic mRNA levels of genes involved in the BA synthesis pathway in Cyp2c70^−/−^ mice and WT littermates on control diet and after 3 weeks of colesevelam treatment. (**H**) mRNA levels of genes involved in BA signaling in the terminal ileum of Cyp2c70^−/−^ mice and WT littermates on control diet and after 3 weeks of colesevelam treatment. N = 8−9 mice/group. Data are presented as Tukey’s box and whisker plots or pie charts, and *p* values represent * *p* < 0.05, ** *p* < 0.01, and *** *p* < 0.001 from Kruskal−Wallis H testing followed by Conover post comparisons. WT: wild type; KO: Cyp2c70^−/−^, WTD: western−type diet; CA: cholic acid; CDCA: chenodeoxycholic acid; Cyp7a1: cholesterol 7α-hydroxylase; Cyp8b1: sterol 12α-hydroxylase; Fxr: farnesoid X receptor; Shp: short heterodimer partner; Fgf15: fibroblast growth factor 15.

**Figure 3 biomedicines-11-02495-f003:**
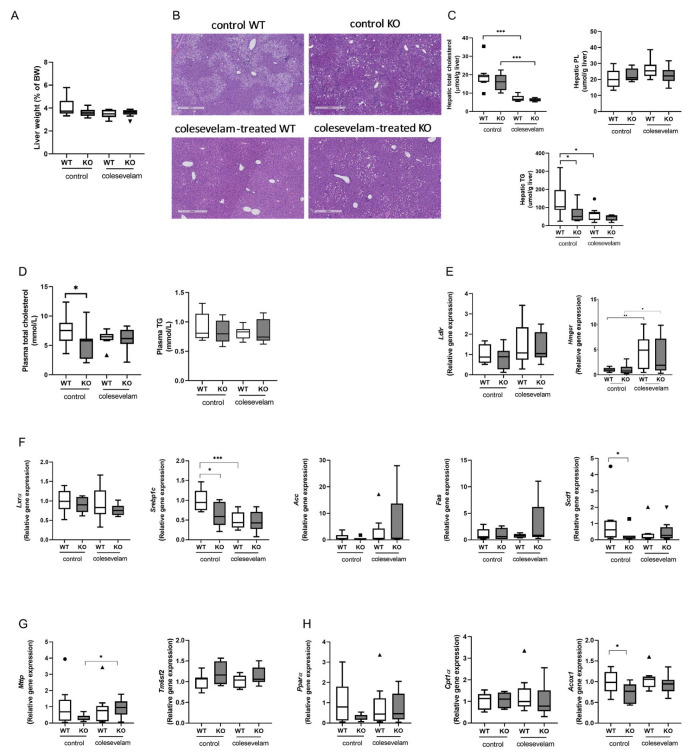
Colesevelam differentially modulates WTD-induced hepatic steatosis in WT and Cyp2c70^−/−^ mice. (**A**) Liver weights of Cyp2c70^−/−^ mice and WT littermates on control diet and after 3 weeks of colesevelam treatment. (**B**) Representative images of H&E-stained liver sections of Cyp2c70^−/−^ mice and WT littermates on control diet and after 3 weeks of colesevelam treatment. (**C**) Hepatic lipid contents (total cholesterol, phospholipids and triglycerides) of Cyp2c70^−/−^ mice and WT littermates on control diet and after 3 weeks of colesevelam treatment. (**D**) Plasma total cholesterol and triglycerides of Cyp2c70^−/−^ mice and WT littermates on control diet and after 3 weeks of colesevelam treatment. (**E**) Hepatic mRNA levels of genes involved in LDL uptake (Ldlr) and cholesterol biosynthesis (Hmgcr) in Cyp2c70^−/−^ mice and WT littermates on control diet and after 3 weeks of colesevelam treatment. (**F**) Hepatic mRNA levels of genes involved in lipogenesis in Cyp2c70^−/−^ mice and WT littermates on control diet and after 3 weeks of colesevelam treatment. (**G**) Hepatic mRNA levels of genes involved in VLDL assembly and secretion in Cyp2c70^−/−^ mice and WT littermates on control diet and after 3 weeks of colesevelam treatment. (**H**) Hepatic mRNA levels of genes involved in fatty acid oxidation (Pparα, Cpt1α, Acox1) in Cyp2c70^−/−^ mice and WT littermates on control diet and after 3 weeks of colesevelam treatment. N = 8–9 mice/group. Quantitative data are presented as Tukey’s box-and−whisker plots and p values represent * *p* < 0.05, ** *p* < 0.01and *** *p* < 0.001 using Kruskal−Wallis H testing followed by Conover post comparisons. PL: phospholipids; TG: triglycerides; Ldlr: low-density lipoprotein receptor; Hmgcr: 3-hydroxy-3-methyl-glutaryl-coenzyme A reductase; Lxrα: liver X receptor alpha; Srebp1c: sterol regulatory element binding protein 1c; Acc: acetyl−CoA carboxylase (Acaca); Fasn: fatty acid synthase; Scd1: acyl−CoA desaturase 1; Mttp: microsomal triglyceride transfer protein; Tm6sf2: transmembrane 6 superfamily member 2; Pparα:peroxisome proliferator−activated receptor alpha; Cpt1α: carnitine palmitoyltransferase 1A; Acox1: acyl−Coenzyme A oxidase 1.

**Figure 4 biomedicines-11-02495-f004:**
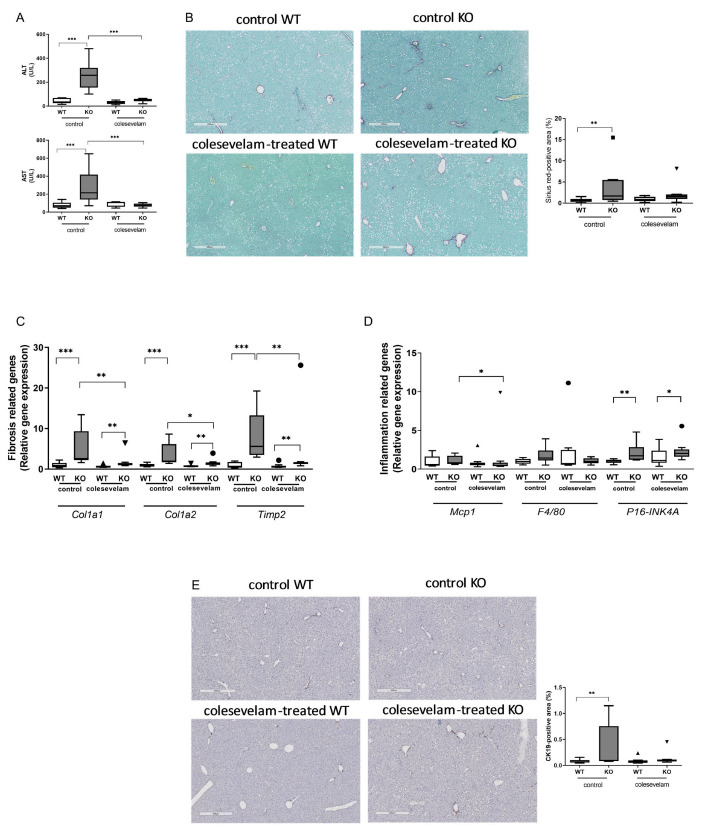
Colesevelam ameliorates liver damage in WTD-fed Cyp2c70^−/−^ mice. (**A**) Levels of the liver damage markers aminotransferase and alanine aminotransferase in plasma of Cyp2c70^−/−^ mice and WT littermates on control diet and after 3 weeks of colesevelam treatment. (**B**) Representative images of SiriusRed Fast Green-stained liver sections and quantification of SiriusRed-positive areas of Cyp2c70^−/−^ mice and WT littermates on control diet and after 3 weeks of colesevelam treatment. (**C**) Hepatic mRNA levels of genes involved in fibrogenesis in Cyp2c70^−/−^ mice and WT littermates on control diet and after 3 weeks of colesevelam treatment. (**D**) Hepatic mRNA levels of genes involved in inflammation and cellular senescence in Cyp2c70^−/−^ mice and WT littermates on control diet and after 3 weeks of colesevelam treatment. (**E**) Representative images of liver sections stained for the cholangiocyte marker CK19 and quantification of CK19-positive areas in Cyp2c70^−/−^ mice and WT littermates on control diet and after 3 weeks of colesevelam treatment. N = 8–9 mice/group. Quantitative data are presented as Tukey’s box-and-whisker plots and p values represent * *p* < 0.05, ** *p* < 0.01, and *** *p* < 0.001 through Kruskal–Wallis H testing followed by Conover post comparisons. ALT: alanine transaminase; AST: aspartate aminotransferase; CK19: cytokeratin 19; Col1a1: collagen type I alpha 1 chain; Col1a2: collagen type I alpha 2 chain; Timp2: metalloproteinase inhibitor 2; Mcp1: monocyte chemoattractant protein 1; F4/80: EGF−like module containing mucin−like hormone receptor−like 1 (Emr1); P16−INK4A: cyclin−dependent kinase inhibitor 2A, (Cdkn2a).

**Figure 5 biomedicines-11-02495-f005:**
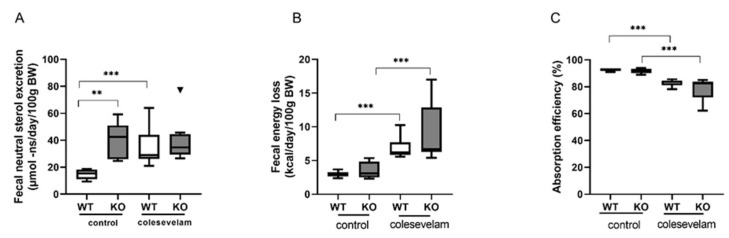
Colesevelam modulates the absorption efficiency of dietary energy in WTD-fed WT and Cyp2c70^−/−^ mice. (**A**) Fecal neutral sterol excretion in Cyp2c70^−/−^ mice and WT littermates on control diet and after 3 weeks of colesevelam treatment. (**B**) Fecal energy content in Cyp2c70^−/−^ mice and WT littermates on control diet and after 3 weeks of colesevelam treatment. (**C**) Absorption efficiency (%) calculated as described in Section 2. N = 8–9 mice/group. Data are presented as Tukey’s box−and−whisker plots and p values represent ** *p* < 0.01 and *** *p* < 0.001 with Kruskal–Wallis H testing followed by Conover post comparisons. NS: neutral sterol.

**Figure 6 biomedicines-11-02495-f006:**
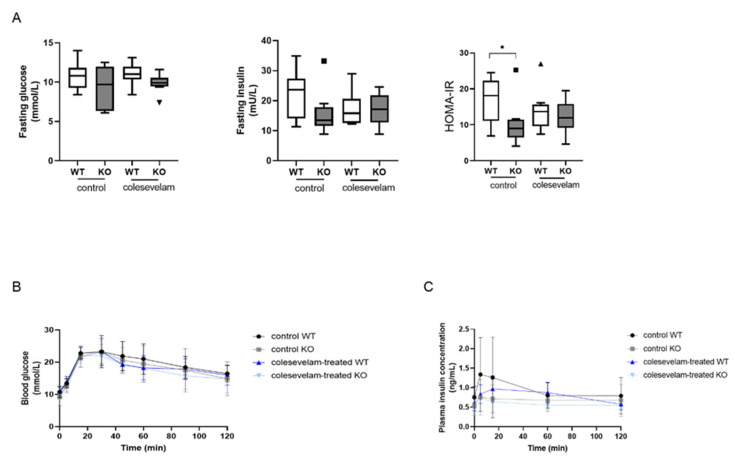
Colesevelam does not affect insulin sensitivity in WTD fed-Cyp2c70^−/−^ mice after 2 weeks of treatment. (**A**) Fasting blood glucose and plasma insulin levels, as well as HOMA-IR in Cyp2c70^−/−^ mice and WT littermates on control diet and after 2 weeks of colesevelam treatment. (**B**) Blood glucose concentrations in Cyp2c70^−/−^ mice and WT littermates during OGTT performed with or without colesevelam treatment for 2 weeks. (**C**) Insulin concentrations in Cyp2c70^−/−^ mice and WT littermates during OGTT performed with or without colesevelam treatment for 2 weeks. N = 8–9 mice/group. Data are presented as mean ± SD and p values represent * *p* < 0.05 with Mann–Whitney U non-parametric comparisons. HOMA−IR: homeostatic model assessment of insulin resistance; OGTT: oral glucose tolerance test.

## Data Availability

All data are in this manuscript.

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
