# Peer review of "Bile Acid Sequestration via Colesevelam Reduces Bile Acid Hydrophobicity and Improves Liver Pathology in Cyp2c70−/− Mice with a Human-like Bile Acid Composition"

_biomedicines, 2023, doi:10.3390/biomedicines11092495_

Round 1
Reviewer 1 Report
The current study examined total cholesterol concentrations in livers of colesevelam-treated WT and KO mice using Cyp2c70-/- mice that are unable to convert chenodeoxycholic acid into rodent-specific muricholic acids (MCAs). Please address the issues listed below.
1. The main goal is stated in the last sentence of the introduction (line 83). However, the reasoning was not made clear.
2. A diet containing 2% (w/w) colesevelam hydrochloride (HCl) that will be influenced by food intake. Please explain how to eliminate this worry.
3. Body composition was measured after 2 weeks of colesevelam treatment, which requires reference(s).
4. How to obtain the plasma or the individual mouse feces for assay? It was not shown in clear.
5. The reference(s) are required for the determination of fecal energy content and energy absorption.
6. The significance of N = 8-9 mice/group is unknown. Please specify the sample size in each case.
7. Colesevelam increased the proportion of 12α-hydroxylated BAs in the circulating bile acid pool in current report. Is it consistent with previous view?
8. KO mice had lower HOMA-IR than WT littermates, which colesevelam did not reverse. Please elaborate on this discovery.
9. The limitations of current findings will be useful.
Author Response
Reviewer n.1
The current study examined total cholesterol concentrations in livers of colesevelam-treated WT and KO mice using Cyp2c70-/- mice that are unable to convert chenodeoxycholic acid into rodent-specific muricholic acids (MCAs). Please address the issues listed below.
- The main goal is stated in the last sentence of the introduction (line 83). However, the reasoning was not made clear.
Preamble: due to their role in several metabolic pathways, bile acids (BAs) and their signalling pathways have been proposed as targets for therapeutic intervention in human (metabolic) diseases. This point is addressed in lines 46-48. It has been observed that BA sequestration reduces plasma LDL cholesterol levels and cardiovascular diseases in patients with hyperlipidaemia and it improves glucose homeostasis in individuals with type 2 diabetes mellitus (T2DM). Point addressed in lines 57-60. Moreover, inhibiting intestinal BA re-absorption alleviates BA-mediated cholestatic liver damage in mouse models of cholestatic liver injury and the BA sequestrant colesevelam improves glycemic control in diabetes murine models (such as db/db mice). The mechanisms underlying these improvements are, as yet, largely unknown (lines 67-68). Importantly, several differences in BA metabolism between mice and humans complicate the translation of murine experimental data to the human situation. Therefore, in our previous studies we generated a mouse model with a more human-like BA composition (Cyp2c70-deficient mice). Western-type diet (WTD)-fed Cyp2c70-/- mice displayed liver injury and with the current study we aimed to determine whether BA sequestration by colesevelam has protective effects on liver pathology and diet-induced insulin resistance in WTD-fed Cyp2c70-/- mice.
- A diet containing 2% (w/w) colesevelam hydrochloride (HCl) that will be influenced by food intake. Please explain how to eliminate this worry.
Individual food intake and body weights were weekly measured. Average food intake during 3 weeks of treatment as well as body weights at the end of treatment were unaffected by colesevelam treatment. This point is addressed in lines 208-210.
- Body composition was measured after 2 weeks of colesevelam treatment, which requires reference(s).
We have previously shown that treating diabetic db/db mice with colesevelam for two weeks improves insulin sensitivity in peripheral tissues (PMID: 22087215 PMCID: PMC3210115 DOI: 10.1371/journal.pone.0024564). This point is addressed in lines 472-474. Based on this, we hypothesised that colesevelam would have an effect already after 2 weeks of treatment. As a result, the body composition as well as the oral glucose tolerance test (OGTT) were performed after two weeks of treatment. Text has been adapted at line 476: “Therefore, we investigated the effects of 2 weeks of colesevelam treatment on glucose metabolism in mice in the context of a human-like BA composition.”
- How to obtain the plasma or the individual mouse feces for assay? It was not shown in clear.
Blood samples were collected after 3 weeks of colesevelam treatment. Mice were fasted for 4 h (08:00–12:00 h) and euthanized, under isoflurane anaesthesia, through cardiac puncture followed by cervical dislocation. Plasma and organs were collected and stored at −80â—¦C until analysis. This point is addressed in lines 103-105.
Fecal samples from each individually-housed mouse were collected at the end of colesevelam treatment for 72h days prior to termination, dried, weighed and thoroughly ground. Text has been adapted at lines 158-160.
- The reference(s) are required for the determination of fecal energy content and energy absorption.
- Li et al., Low production of 12α-hydroxylated bile acids prevents hepatic steatosis in Cyp2c70-/- mice by reducing fat absorption, J Lipid Res, vol. 62, 2021, doi: 10.1016/J.JLR.2021.100134. Reference added at line 194.
- The significance of N = 8-9 mice/group is unknown. Please specify the sample size in each case.
Total Cyp2c70-/- (KO) mice: n=17.
Total wild-type (WT) littermates: n=18.
9 KO mice and 9 WT mice were subsequently treated with 2% (w/w) colesevelam hydrochloride (HCl).
8 KO mice and 9 WT mice remained on WTD without colesevelam as control.
This point is addressed in lines 89-90/96-100.
- Colesevelam increased the proportion of 12α-hydroxylated BAs in the circulating bile acid pool in current report. Is it consistent with previous view?
It is in line with our previous findings:
We have previously shown that a low production of 12α-hydroxylated bile acids (CA, DCA and their tauro-coniugated forms) prevents hepatic steatosis in Cyp2c70-/-. In the current study we show that colesevelam reduces the hydrophobicity of the bile acid pool in Cyp2c70-/- mice due to an increase in proportion of CA at the expense of the hydrophobic CDCA. This reduction in hydrophobicity of biliary bile acids could likely contributes to the amelioration of liver pathology in Cyp2c70-/- mice.
Furthermore, this increase of 12α-hydroxylated BAs is also supported by Truong et al. They show that the interruption of the enterohepatic circulation of BAs by pharmacological inhibition of the ileal BA transporter (IBAT/ASBT/SLC10A2) reduced the proportion of (T)CDCA and increased the proportion of (T)CA) plus (T)DCA. (see reference [30] in manuscript)
- KO mice had lower HOMA-IR than WT littermates, which colesevelam did not reverse. Please elaborate on this discovery.
Although HOMA-IR was lower in Cyp2c70-/- mice compared to their WT littermates, treatment with colesevelam for 2 weeks had no effects on HOMA-IR in either group and it had no effects on glucose excursions or plasma insulin levels during the OGTT.
Based on these findings we hypothesize that the unobserved improvement in glucose homeostasis and insulin sensitivity in our Cyp2c70-/- mice could be due to a high sequestration of hydrophobic CDCA by colesevelam in Cyp2c70-/- mice. Since more hydrophobic BAs are potent agonists for TGR5 receptor (its activation leads to an improvement of the whole-body glucose homeostasis and insulin sensitivity by promotion of GLP-1 secretion), reduced CDCA levels could not effectively activate this receptor and, therefore, may not lead to an improvement of glucose homeostasis. Additionally, decreased CDCA levels may interfere with AKT phosphorylation and lower muscle insulin sensitivity. (see reference [32] in manuscript)
- The limitations of current findings will be useful.
Although we show that colesevelam has no effects on glucose homeostasis and insulin sensitivity, one limitation of our findings is represented by the considerably shorter duration of colesevelam treatment (3 weeks) compared to the 8 weeks of ASBT inhibition applied by Truong and colleagues.
Furthermore, it may be useful to study the effects of colesevelam in mice with a human-like BA composition in the context of a more severe insulin resistant phenotype at the start of the intervention (such as db/db mice). Addressed in lines 490-493.
Reviewer 2 Report
In this manuscript, the authors examined the effect of the bile acid (BA)sequestrant colesevelam on BA metabolism and diet-induced steatosis and liver damage in Cyp2c70-/- (KO) mice, a mouse model with a human-like BA composition. Colesevelam treatment increased BA excretion and reduced the hydrophobicity of biliary BAs in KO mice. KO mice were resistant to diet-induced steatosis, while wild-type (WT) mice showed diet-induced steatosis, which was ameliorated by colesevelam. KO mice had increased liver damage and increased expression of fibrogenesis-related genes compared to WT mice. These phonotypes of KO mice were suppressed by colesevelam treatment. HOMA-IR was decreased in KO mice, but colesevelam was not effective on glucose metabolism in WT or KO mice. The results provide insight into the role of human bile acids in liver disease.
1. Fig. 2. BA levels in feces and plasma were examined, and BA composition in gallbladder was analyzed. How about total BA levels in gallbladder (or BA pool).
2. Fig. 2. The changes in BA hydrophobicity in WT and KO mice with/without colesevelam may be related to the Cyp7a1 and Cyp8b1 expression. Please discuss the mechanism of the changes of BA hydrophobicity in more detail. The patterns of Cyp7a1 and Cyp8b1 expression and different from those of Shp and intestinal Fgf15. Can the patterns of Cyp7a1 and Cyp8b1 be explained by FXR activation? Please discuss what factor(s) regulate Cyp7a1 and Cyp8b1 expression in more detail.
3. BA levels and composition were not examined in WT or KO mice fed a Western type-diet. Is there any difference when compared to those in mice fed a control diet?
4. Fig. 3. Colesevelam increased Hmgcr expression but not ldlr expression in WT and KO mice. Please discuss possible mechanism(s).
5. Please discuss why KO mice are resistant to diet-induced steatosis in more detail by comparing the results reported in other reports.
6. Related to the above comment. Do human bile acids, such as CDCA and TCDCA, suppress steatosis and induce liver damage (inflammation/fibrosis)? Is the BA hydrophobicity related to the pathogenesis of steatosis and liver damage? The differences in the effects of colesevelam and ASMT inhibitors are discussed well. Please discuss the underlying mechanisms for the observed findings in more detail.
Author Response
Reviewer n.2
In this manuscript, the authors examined the effect of the bile acid (BA)sequestrant colesevelam on BA metabolism and diet-induced steatosis and liver damage in Cyp2c70-/- (KO) mice, a mouse model with a human-like BA composition. Colesevelam treatment increased BA excretion and reduced the hydrophobicity of biliary BAs in KO mice. KO mice were resistant to diet-induced steatosis, while wild-type (WT) mice showed diet-induced steatosis, which was ameliorated by colesevelam. KO mice had increased liver damage and increased expression of fibrogenesis-related genes compared to WT mice. These phonotypes of KO mice were suppressed by colesevelam treatment. HOMA-IR was decreased in KO mice, but colesevelam was not effective on glucose metabolism in WT or KO mice. The results provide insight into the role of human bile acids in liver disease.
- 2. BA levels in feces and plasma were examined, and BA composition in gallbladder was analysed. How about total BA levels in gallbladder (or BA pool).
Total BA concentrations in gallbladder bile were not quantitative since the gallbladders and the volume of bile were too small for a precise and quantitative analysis. However, the percentages were calculated, accurate and used to calculate the hydrophobicity index. Text has been adapted at lines 141-144.
The BA pool includes the total BAs circulating in the enterohepatic circulation, including BAs in the liver, intestine and gallbladder. We did not measure the levels of BA in the intestine; therefore, the total pool size was not indicated.
- 2. The changes in BA hydrophobicity in WT and KO mice with/without colesevelam may be related to the Cyp7a1 and Cyp8b1 expression. Please discuss the mechanism of the changes of BA hydrophobicity in more detail. The patterns of Cyp7a1 and Cyp8b1 expression and different from those of Shp and intestinal Fgf15. Can the patterns of Cyp7a1 and Cyp8b1 be explained by FXR activation? Please discuss what factor(s) regulate Cyp7a1 and Cyp8b1 expression in more detail.
Western type diet-fed Cyp2c70-/- mice upon colesevelam showed an increased in 12α-hydroxylated BA (cholic acid, deoxycholic acid, and their conjugated forms) at the expenses of non 12α-hydroxylated BA. This lead to a significantly decreased hydrophobicity of biliary BAs. In line with this shift towards more 12α-hydroxylated CA and less hydrophobic CDCA, Cyp2c70-/- mice showed an increased hepatic mRNA expression of Cyp8b1, encoding sterol 12α-hydroxylase and therefore responsible for CA synthesis.
Colesevelam, being a BA sequestrant it binds BA in the intestine leading to the formation of non-absorbable complexes which are excreted with the feces. This increases loss of BA in faeces. In line with this, colesevelam significantly increased faecal BA excretion in both WT and Cyp2c70-/- mice, reflecting the sequestration action of colesevelam. In order to compensate this fecal BA loss, BA synthesis was stimulated. In facts, hepatic mRNA levels of Cyp7a1, the rate limiting enzyme in the classical BA synthesis pathway, were strongly upregulated in colesevelam-treated mice compared to untreated controls.
The nuclear receptor FXR, once activated, regulates Cyp8b1 and Cyp7a1 by inhibiting their expression. Upon colesevelam, the sequestration of CDCA, a potent endogenous FXR activator, may not efficiently activate FXR. However, no significant changes of hepatic Fxr expression were observed in KO mice upon colesevelam treatment. In line with the prevention of intestinal BA uptake upon colesevelam, the expression of the FXR downstream target genes Shp and Fgf15 was reduced in both WT and KO mice. (This information is presented in the manuscript in lines 234-238 and 240-245).
- BA levels and composition were not examined in WT or KO mice fed a Western type-diet. Is there any difference when compared to those in mice fed a control diet?
BA levels in WT and KO mice on WTD (Western type-diet) were measured and they are named as “WT and KO on control diet”. Our control diet is represented by the WTD. This point is addressed in lines 99-100
- 3. Colesevelam increased Hmgcr expression but not ldlr expression in WT and KO mice. Please discuss possible mechanism(s).
Indeed, there was a significant increase in hepatic mRNA levels of the gene involved in cholesterol biosynthesis (Hmgcr) upon colesevelam treatment in both WT and KO mice compared to untreated counterparts. This is in line with the action of BA sequestrant: in order to compensate the BA loss, BA synthesis was stimulated. The precursor of BAs needed for their synthesis is cholesterol. Therefore, our results indicated that the sequestration of BA induced cholesterol synthesis to provide substrate for BA synthesis. Surprisingly, colesevelam treatment had no impact on the hepatic mRNA levels of LDL receptor. Taken together these results suggest that cholesterol synthesis, and not the uptake of cholesterol-carrying lipoprotein particles, is stimulated by colesevelam. The mechanisms underline this absence of effects on ldlr expression, however, remain to be elucidated. We think that elaboration of this issue is beyond the scope of the current work but if the reviewer insist we are willing to incorporate this in the discussion.
- Please discuss why KO mice are resistant to diet-induced steatosis in more detail by comparing the results reported in other reports.
The current study shows the presence of diet-induced hepatic steatosis in livers of WTD-fed WT mice and to a lesser extend in WTD-fed KO mice. We have previously demonstrated that Cyp2c70-/- mice are protected from WTD-induced obesity (especially females) and hepatic steatosis in both genders, primarily due to impaired intestinal fat absorption (See reference [22] in this manuscript) PMID: 34626589).
- Related to the above comment. Do human bile acids, such as CDCA and TCDCA, suppress steatosis and induce liver damage (inflammation/fibrosis)? Is the BA hydrophobicity related to the pathogenesis of steatosis and liver damage? The differences in the effects of colesevelam and ASBT inhibitors are discussed well. Please discuss the underlying mechanisms for the observed findings in more detail.
It has been reported that the hydrophobic CDCA acid use in humans is limited due to negative side effects: diarrhea, elevated levels of palsma transaminases and it can cause (mild) hepatotoxicity in some subjects (PMID: 26233910). It is also known that BA hydrophobicity is accompanied with cytotoxicity: the more hydrophobic BAs are, the more toxic they are to cells.
Regarding the BA hydrophobicity and its link to liver phenotype in our model, our previous findings reveal that (especially) chow-fed female Cyp2c70-/- mice display features of cholangiopathy that progresses to hepatic fibrosis as they age. Intriguingly, treating female Cyp2c70-/- mice with hydrophilic UDCA was able to completely reverse these pathological features, indicating that BA composition and thus hydrophobicity play an important role in liver phenotype (see reference [22] of this manuscript, PMID: 33309945). Hence, BA hydrophobicity is linked to liver damage in Cyp2c70-/- mice.
In the current study we show that male WTD-fed Cyp2c70-/- mice display features of liver damage, which were attenuated by BA sequestration via colesevelam, able to decrease the hydrophobicity of biliary BAs in Cyp2c70-/- mice.
To conclude, the reason why Cyp2c70-/- mice exhibit liver damage but humans with a comparable hydrophobicity index of biliary BAs do not, can be explained by the fact that these mice cannot tolerate the hydrophobicity condition generated by the absence of hydrophilic MCAs.
This issue is addressed in lines 403-412 in the manuscript.
Reviewer 3 Report
Opinion about the manuscript entitled „Bile acid sequestration by colesevelam reduces bile acid hydrophobicity and improves liver pathology in Cyp2c70-/- mice with a human-like bile acid composition” sent to Biomedicines (MDPI).
I appreciate your work and after minor explanations the text could be published in a respectable journal Biomedicines.
Comments and queries:
You stated that your mice are characterized with development of liver fibrosis. What stage of fibrosis they developed? A normal liver is at a stage between F0 and F1. Stage F2 denotes light fibrosis, and F3 is severe fibrosis. ‘Cirrhosis’ is defined from stage F4, when scar tissue exists throughout the liver. Please add an appropriate information to the text. Additionally, in a small number of cases, steatosis can develop into a fibrosis that can lead to cirrhosis. What is the health status of the liver in your untreated mice? Could a developed fibrosis cover some diet-induced pathologies that can be seen in “normal” WT mice?
Hydrophobic bile acids, including glycocholic acid (GCA), cholic acid (CA), lithocholic acid (LCA), chenodeoxycholic acid (CDCA), and deoxycholic acid (DCA) are a major factor in inducing liver cell death. That is scientific truth. According to your studies are you sure that a simple decrease in one of those acids in favour for hydrophilic MCA can help the liver in visible therapeutic manner (apart from colesevelam)?
The graphical visualization of data is correct and readable. The language is understandable. The references are used in a proper manner without any bias.
The conclusions could be extended, for instance with suggestions why BA sequestration had no effects on insulin level? A longer treatments could have effects?
Author Response
Reviewer n.3
Opinion about the manuscript entitled “Bile acid sequestration by colesevelam reduces bile acid hydrophobicity and improves liver pathology in Cyp2c70-/- mice with a human-like bile acid composition” sent to Biomedicines (MDPI).
I appreciate your work and after minor explanations the text could be published in a respectable journal Biomedicines.
Comments and queries:
- You stated that your mice are characterized with development of liver fibrosis. What stage of fibrosis they developed? A normal liver is at a stage between F0 and F1. Stage F2 denotes light fibrosis, and F3 is severe fibrosis. ‘Cirrhosis’ is defined from stage F4, when scar tissue exists throughout the liver. Please add an appropriate information to the text. Additionally, in a small number of cases, steatosis can develop into a fibrosis that can lead to cirrhosis. What is the health status of the liver in your untreated mice? Could a developed fibrosis cover some diet-induced pathologies that can be seen in “normal” WT mice?
According to our previous studies, livers of untreated WTD-fed KO mice displayed moderately increased collagen deposition compared to their WT littermates, which was mostly restricted to the periportal regions. It is worth noting that the degree of fibrosis varied within the group of untreated KO mice. à F2 in KO mice compared to a WT liver (F0/F1). Text has been adapted at lines 326-327.
We cannot rule out the possibility that a developed fibrosis might cover some diet-induced pathologies. In fact, in another previous study, we show that, despite the presence of hepatic fibrosis, especially in female Cyp2c70-/- mice, female Cyp2c70-/- mice are protected from WTD-induced obesity, insulin resistance, and hepatic steatosis. (See reference [22] in current manuscript)
- Hydrophobic bile acids, including glycocholic acid (GCA), cholic acid (CA), lithocholic acid (LCA), chenodeoxycholic acid (CDCA), and deoxycholic acid (DCA) are a major factor in inducing liver cell death. That is scientific truth. According to your studies are you sure that a simple decrease in one of those acids in favour for hydrophilic MCA can help the liver in visible therapeutic manner (apart from colesevelam)?
Based on our (previous) results, the high hydrophobicity of the BA pool in Cyp2c70-/- mice (due to the absence of hydrophilic MCAs and to abundance of hydrophobic CDCA) plays an important role in the development of liver damage (PMID: 33309945). Livers of Cyp2c70-/- mice cannot tolerate this hydrophobicity condition. Supporting the importance of hydrophobicity on liver phenotype, we have already demonstrated that increasing the hydrophilicity with (hydrophilic) UDCA fully restores the cholangiopathy in Cyp2c70-/- mice (PMID: 33309945). In the current study we show that BA sequestration by colesevelam significantly decreased the hydrophobicity of biliary BAs in Western-type diet (WTD)-fed Cyp2c70-/- mice, by increasing the ratio of 12α-hydroxylated BA to non-12α-hydroxylated BAs. This shift in BA production towards more CA and less CDCA may partially contribute to the beneficial effects of the BA-sequestrant colesevelam on liver pathology in Cyp2c70-/- mice, leading to a more hydrophilic, less cytotoxic BA composition in Cyp2c70-/- mice.
Our results emphasise the important role of BA composition and hydrophobicity on liver phenotype.
- The graphical visualization of data is correct and readable. The language is understandable. The references are used in a proper manner without any bias.
We thank the reviewer for this positive comment
- The conclusions could be extended, for instance with suggestions why BA sequestration had no effects on insulin level? A longer treatment could have effects?
It can be hypothesized that the unobserved effects (improvement) on glucose homeostasis and insulin sensitivity in our Cyp2c70-/- mice could be due to a high sequestration of hydrophobic CDCA by colesevelam. It is known that more hydrophobic BAs are potent agonists for TGR5 receptor: its activation leads to an improvement of the whole-body glucose homeostasis and insulin sensitivity by promotion of GLP-1 secretion. Therefore, reduced CDCA levels could not effectively activate this receptor and, hence, may not lead to an improvement of glucose homeostasis. Additionally, decreased CDCA levels may interfere with AKT phosphorylation and lower muscle insulin sensitivity. Rather than a longer treatment, it would be more useful to further investigate the effects of colesevelam on a more sever insulin resistance phenotype (such as db/db mice) prior the intervention. Text has been adapted at lines 482-490.
Round 2
Reviewer 2 Report
The authors responded to the reviewers' comments appropriately. However, only some of the responses are reflected in the revised manuscript. Please revise the manuscript by including the information in the responses.
Author Response
Comments and Suggestions for Authors
The authors responded to the reviewers' comments appropriately. However, only some of the responses are reflected in the revised manuscript. Please revise the manuscript by including the information in the responses.
Response from the author:
Added at lines 141-144: Total BA concentrations in gallbladder bile were not quantitated since the gallbladder volumes were too small for a precise and quantitative analysis. However, the BA percentages were calculated, accurate and used to calculate the hydrophobicity index – answer to question n1
Added at lines 219-221: Colesevelam, being a BA sequestrant, binds BA in the intestine leading to the formation of non-absorbable complexes that are excreted into the feces and thereby increases fecal BA loss. In line with this, (BA sequestration significantly increased fecal bile acid excretion in both WT and KO mice) - answer to question n2
Added at lines 240, 241: Furthermore, to compensate for the colesevelam-induced fecal BA loss, BA synthesis was stimulated. In fact, … - answer to question n2
Added at line 244, 245: (Cyp7a1, considered the rate-limiting enzyme in bile acid synthesis, and Cyp8b1, controlling for CA synthesis, were significantly increased in colesevelam-treated mice compared to untreated controls) in line with the shift towards more 12α-hydroxylated (T)CA and less hydrophobic (T) CDCA - answer to question n2
Added at lines 246,248: The nuclear receptor FXR, once activated, regulates Cyp8b1 and Cyp7a1 by inhibiting their expression. Upon colesevelam treatment, sequestration of CDCA, a potent endogenous FXR activator, activation of FXR may be less effective. (However, no significant changes of hepatic) - answer to question n2
Added at line 292: treatment with the BA sequestrant indeed induced cholesterol synthesis to provide substrate (cholesterol) for BA production. - answer to question n4
Added 292,295: Taken together these results suggest that cholesterol synthesis, and not the uptake of cholesterol-carrying lipoprotein particles, is stimulated by colesevelam. The mechanisms underlying the absence of effects on Ldlr expression, however, remain to be elucidated. - answer to question n4